# VARIABLE-SHOT ADAPTATION FOR ONLINE META-LEARNING

## ABSTRACT

Few-shot meta-learning methods consider the problem of learning new tasks from a small, fixed number of examples, by meta-learning across static data from a set of previous tasks. However, in many real world settings, it is more natural to view the problem as one of minimizing the total amount of supervision — both the number of examples needed to learn a new task and the amount of data needed for meta-learning. Such a formulation can be studied in a sequential learning setting, where tasks are presented in sequence. When studying meta-learning in this online setting, a critical question arises: can meta-learning improve over the sample complexity and regret of standard empirical risk minimization methods, when considering both meta-training and adaptation together? The answer is particularly non-obvious for meta-learning algorithms with complex bi-level optimizations that may demand large amounts of meta-training data. To answer this question, we extend previous meta-learning algorithms to handle the variable-shot settings that naturally arise in sequential learning: from many-shot learning at the start, to zero-shot learning towards the end. On sequential learning problems, we find that meta-learning solves the full task set with fewer overall labels and achieves greater cumulative performance, compared to standard supervised methods. These results suggest that meta-learning is an important ingredient for building learning systems that continuously learn and improve over a sequence of problems.

## 1 INTRODUCTION

Standard machine learning methods typically consider a static training set, with a discrete training phase and test phase. However, in the real world, this process is almost always cyclical: machine learning systems might be improved with the acquisition of new data, repurposed for new tasks via finetuning, or might simply need to be adjusted to suit the needs of a changing, non-stationary world. Indeed, the real world is arguably so complex that, for all practical purposes, learning is never truly finished, and any real system in open-world settings will need to improve and finetune perpetually (Chen & Asch, 2017; Zhao et al., 2019). In this continual learning process, meta-learning provides the appealing prospect of accelerating how quickly new tasks can be acquired using past experience, which in principle should make the learning system more and more efficient over the course of its lifetime. However, current meta-learning methods are typically concerned with asymptotic few-shot performance (Finn et al., 2017; Snell et al., 2017). For a continual learning system of this sort, we instead need a method that can minimize both the number of examples per task, and the number of tasks needed to accelerate the learning process.

Few-shot meta-learning algorithms aim to learn the structure that underlies data coming from a set of related tasks, and use this structure to learn new tasks with only a few datapoints. While these algorithms enable efficient learning for new tasks at test time, it is not clear if these efficiency gains persist in online learning settings, where the efficiency of both meta-training and few-shot adaptation is critical. Indeed, simply training a model on all received data, i.e. standard supervised learning with empirical risk minimization, is a strong competitor since supervised learning methods are known to generalize well to in-distribution tasks in a zero-shot manner. Moreover, it's not clear that meta-learning algorithms can improve over such methods by leveraging shared task structure in online learning settings. Provided that it is possible for a single model to fully master all of the tasks with enough data, both meta-learning and standard empirical risk minimization approaches should produce a model of equal competence. However, the key hypothesis of this work is that meta-learned

Figure 1: **Online incremental meta-learning.** We visualize our online incremental meta-learning problem setting in the above figure using the Incremental Pose Prediction dataset discussed in Section 7. At round t, the current task is to predict the pose of a sofa with a small training set containing several datapoints of the sofa with different orientations. After one epoch of training, we evaluate the few-shot generalization performance on a test datapoint where the number of shots equals the size the task's training set. If the test performance exceeds some proficiency threshold C, we advance to the next task, i.e. predicting the pose of the airplane. Otherwise, we add another training example of the sofa to the training set and repeat the above process.

models will become more accurate more quickly in the middle of the online learning process, while data is still being collected, resulting in lower overall regret in realistic problem settings.

To test this hypothesis, we consider a practical online learning problem setting, which we refer to as online incremental learning, where the algorithm must learn a sequence of tasks, and datapoints from each task are received sequentially. Once a model reaches a certain level of proficiency, the algorithm may move on to training the model on the next task. This problem is crucial to solve in the real world, especially in settings where online data collection and supervision signals are costly to obtain. One example of such a problem definition is a setting where a company receives requests for object classifiers, sequentially at different points in time. Data collection and labels are expensive, and the company wants to spend the least amount of money on acquiring a good classifier for each request. A major challenge for meta-learning that arises in this problem setting is to design a meta-learning algorithm that can generalize with variable shots: As data from new tasks is incrementally introduced, at any given point in time, the model may have access to zero, a few, or many datapoints for a provided task. The goal of this work is to achieve variable-shot adaptation while minimizing the total amount of supervision in terms of number of shots required for each added task. The desired online incremental meta-learning algorithm is expected to generalize to a new task with decreasing number of shots over the course of training. We visualize our problem setting in Figure 1.

The key contributions of this work are (a) a new meta-learning algorithm that can adapt to variable amounts of data, and (b) an online version of this algorithm that addresses the above problem setting of online incremental learning. We theoretically derive our variable-shot meta-learning algorithm and combine it with deep neural networks for effective online learning on challenging sequential problem settings. Perhaps surprisingly, we find that our approach can outperform empirical risk minimization and a previous online meta-learning method (Finn et al., 2019) on two online image classification problems consisting of sequences of classification tasks and one online regression problem. Further, we find that, in the offline setting, our approach performs comparably to previous state-of-the-art algorithms in few-shot learning, and provides considerable gains in the variable-shot setting.

## 2 PRELIMINARIES

Meta-learning algorithms optimize for efficient adaptation to new tasks. To define the meta-learning problem, let $p(\mathcal{T})$ denote a *task distribution*, where each task $\mathcal{T}_i \sim p(\mathcal{T})$ consists of a dataset $\mathcal{D}_i := \{\mathbf{x}, \mathbf{y}\}$ with i.i.d. input and output pairs. If we have a predictive model $\mathbf{h}(\mathbf{x}; \theta)$ with some parameter $\theta$ and a loss function $\ell$, such as the cross entropy between the predicted label distribution and the true label distribution in a classification problem, the risk of $\mathcal{T}_i$, $f_i$, can be computed as $f_i(\theta) = \mathbb{E}_{(\mathbf{x},\mathbf{y}) \sim \mathcal{D}_i} [\ell(\mathbf{h}(\mathbf{x}; \theta), \mathbf{y})]$. At meta-training time, $N$ tasks $\{\mathcal{T}_i\}_{i=1}^N$ are sampled from $p(\mathcal{T})$ and meta-learning algorithms aim to learn how to quickly learn these tasks such that, at meta-test time, the population risk $f_j(\theta)$ of the unseen tasks $\mathcal{T}_j \sim p(\mathcal{T})$ is minimized quickly.

In this work, we build on the model-agnostic meta-learning (MAML) algorithm (Finn et al., 2017). MAML achieves fast adaptation to new tasks by optimizing a set of initial parameters $\theta_{\text{MAML}}$ that can be quickly adapted to the meta-training tasks $\{\mathcal{T}_i\}_{i=1}^N$. Thus, at meta-test time, after a small number of gradient steps on $\theta_{\text{MAML}}$ with $K$ datapoints from $\mathcal{D}_j$, the model can minimize $f_j$ for the new task $\mathcal{T}_j$. Note that $K$ is a small and fixed number across all tasks. Formally, MAML achieves such an initialization by optimizing the following objective: $\min_{\theta_{\text{MAML}}} \frac{1}{N} \sum_{i=1}^N f_i(\mathbf{U}_i(\theta_{\text{MAML}}, \alpha, K_i))$ where

$\alpha$ is the inner gradient step size, $\mathbf{U}_i(\theta, \alpha, K_i) = \theta - \alpha \nabla_\theta \hat{f}_i^K(\theta)$ denotes the gradient update in the inner loop for task $i$, and $\hat{f}_i^K(\theta) = \frac{1}{K} \sum_{j=1}^K \ell(\mathbf{h}(\mathbf{x}_j; \theta), \mathbf{y}_j)$ is the empirical risk of $\mathcal{T}_i$ based on a minibatch with $K$ datapoints sampled from $\mathcal{D}_i$. Hence, MAML can be viewed as optimizing for K-shot generalization.

In the online setting, prior work (Finn et al., 2019) has proposed the follow-the-meta-leader (FTML) algorithm, which adapts MAML to an online meta-learning setting where the meta-learner learns to adapt to tasks that arrive in a sequence. In this online meta-learning setting, the meta-learner faces a sequence of loss functions $\{\ell_t\}_{t=1}^\infty$ in each round $t$, where each of the tasks share some common structure that could be captured by the meta-learner. The goal of the meta-learner is to find a sequence of models parametrized by $\{\theta_t\}_{t=1}^\infty$ that performs well on the sequence of loss functions after making some task-specific update procedures $\mathbf{U}_t(\theta_t, \alpha, K) : \theta_t \in \Theta, K \in \mathbb{N} \to \theta' \in \Theta$ where $K$ is a fixed number of datapoints used to perform the update. In FTML, which adapts MAML to the online setting, $\mathbf{U}_t(\theta_t, \alpha, K)$, as defined in the paragraph above, is the inner gradient update w.r.t. the empirical risk with $K$-shot data $\hat{f}_t^K$ at round $t$. The performance of FTML is typically measured using the regret that compares the sum of population risks to the cumulative population risks with some fixed model parameter computed in hindsight: $\text{Regret}_T = \sum_{t=1}^T f_t(\mathbf{U}_t(\theta_t, \alpha, K)) - \min_\theta \sum_{t=1}^T f_t(\mathbf{U}_t(\theta, \alpha, K))$. Analogously to the standard follow-the-leader (FTL) algorithm (Hannan, 1957; Kalai & Vempala, 2005), FTML optimizes $\theta_t$ as $\theta_{t+1} = \arg\min_\theta \sum_{j=1}^t f_j(\mathbf{U}_j(\theta, \alpha, K))$. As mentioned above, the number of datapoints used for computing updates, $K$, is always fixed in both offline and online meta-learning settings. However, in the sequential learning setting, where datapoints for each task arrive incrementally, the model is required to handle varying numbers of datapoints from different tasks, and hence needs to generalize to new tasks quickly using *any* amount of data. Our method is designed to tackle this problem setting, which we will discuss in the next section.

## 3 ONLINE INCREMENTAL META-LEARNING PROBLEM STATEMENT

The online incremental meta-learning problem statement extends the online meta-learning setting (Finn et al., 2019), where the tasks arrive one at a time. In our proposed incremental setting, the data *within* each task also arrives one data point at a time, and the goal of the model is to minimize the cumulative regret summed over all of the tasks. A model that can adapt to a new task more quickly in fewer shots should attain lower regret. A model that can do so *while having seen as few previous tasks as possible* will be even better. Formally, the learning process is divided into "tasks" and "shots". Each time step, the learner either receives one more "shot" (data point) for the current task, or transitions to a new task and receives the first shot for that task. In practice, instead of generating 1 datapoint every time step, the learner receives a small batch of datapoints every few time steps. Let $t$ denote the current task counter, and $s$ the current shot counter. At each step $s$ for task $t$, the model has access to $\hat{\mathcal{D}}_t(s)$, the dataset for the current task, which presently contains $s$ shots. The model can adapt to the current task using $\hat{\mathcal{D}}_t(s)$, and must then attain the lowest possible risk on that task.

To realize this setting, we extend the update procedure defined in Section 2 to the variable-shot setting, formally $\mathbf{U}_t(\theta_t, \alpha, s)$. The risk in round $t$ with $s$ shots in this incremental setting is defined as $f_t(\mathbf{U}_t(\theta_t, \alpha, \min\{s, M\}))$ where $M$ is some large scalar representing the maximum number of shots per task. The risk defined above suggests that the performance of the model is only evaluated using the $s$-shot risk $f_t(\mathbf{U}_t(\theta_t, \alpha, s))$, when there are $s <= M$ datapoints in the current datasets, and always use $M$-shot risk when there are more than $M$ datapoints.

Now that we have defined our measure of incremental risk, we will discuss when the model can transition from one task to the next. Intuitively, we want the model to be able to "complete" a task and transition to a new task as soon as it has achieved a requisite accuracy on that task. For instance, in the example in Section 1 of the company that receives requests for object classifiers, the company might choose to stop collecting training data for a given task once accuracy on that task is good enough, and move on to the next customer. We formalize this by introducing a threshold value $C$, such that when $f_t(\mathbf{U}_t(\theta_t, \alpha, s)) \leq C$ at step $s$, the learner switches to the next task immediately. In this way, a very efficient model that can adapt and master a task in just a few data points will attain lower overall regret and solve the full sequence of tasks with fewer overall datapoints, since it will need fewer shots for each task.

To summarize, the overall online incremental meta-learning process looks like this: **(1)** At the start of round $t$, the meta-learner selects a model parameterized by parameter $\theta_t$; **(2)** The world chooses a

task $\mathcal{T}_t \sim p(\mathcal{T})$; **(3)** At each step $s = 0, \ldots$, the dataset $\hat{\mathcal{D}}_t(s)$ receives one datapoint; **(4)** At step $s = 0, \ldots$, the model updates $\theta_t$ as $\theta_t' = \mathbf{U}_t(\theta_t, \alpha, s)$; **(5)** At step $s = 0, \ldots$, the model is evaluated with $f_t(\mathbf{U}_t(\theta_t, \alpha, s))$ and if $f_t(\mathbf{U}_t(\theta_t, \alpha, s)) \leq C$, the agent will advance to the next round. When the advancement happens, denote $S_t = s$. The objective of incremental meta-learning is to minimize the regret over all the rounds, resembling the objective of online learning (Hannan, 1957; Kalai & Vempala, 2005), which we state as follows:

$$\text{Regret}_T = \sum_{t=1}^{T} \sum_{s=0}^{S_t} f_t(\mathbf{U}_t(\theta_t, \alpha, \min\{s, M\})) - \min_\theta \sum_{t=1}^{T} \sum_{s=0}^{S_t} f_t(\mathbf{U}_t(\theta, \alpha, \min\{s, M\})),$$

where $\{\theta_t\}_{t=1}^{T}$ is a sequence of models at round $t = 1, \ldots, T$. Similar to FTML, we aim to obtain a meta-learner that learns each task in the data-incremental setting, while attaining minimal regret.

## 4 VARIABLE-SHOT MODEL-AGNOSTIC META-LEARNING

The first step toward designing an online incremental meta-learning algorithm that can address this problem statement is to design a method that can handle variable numbers of shots, since in the incremental setting, the model will need to accommodate settings ranging from zero-shot at the beginning of each task to many-shot at the end. Attaining the lowest regret therefore depends crucially on maximally utilizing whatever task data may be available. We will therefore first design a variable-shot generalization of MAML in the standard *offline* setting, and then apply it to the *online* setting to provide a viable approach for online incremental meta-learning.

### 4.1 A NAÏVE SOLUTION

A naive solution would simply involve applying MAML with varying numbers of datapoints used to compute the inner graduate updates. During meta-training, the number of datapoints could be drawn uniformly from $\{0, \ldots, M\}$ for some $M \in \mathbb{Z}^+$. The corresponding meta-training objective is then given as $\min_\theta \frac{1}{N} \sum_{i=1}^{N} \mathbb{E}_{s \sim \text{Unif}(0, M)} \left[ f_i(\theta - \alpha \nabla_\theta \hat{f}_i^s(\theta)) \right]$.

Unfortunately, as we will see in Section 7, this approach does not perform well when the number of shots varies, since the adaptation process is not aware of the number of shots that are available. Intuitively, when more data is available, the model should be able to deviate further from the prior parameter vector $\theta$, since the larger dataset provides us with more information about the task. Mathematically, this can be captured by the variance of the inner gradient. The variance of the inner gradient depends on $s$ as follows: $\text{Var}(\nabla_\theta \hat{f}_i^s(\theta)) = \text{Var}\left(\nabla_\theta \frac{1}{s} \sum_{j=1}^{s} \ell(\mathbf{h}(\mathbf{x}_j; \theta), \mathbf{y}_j)\right) = \frac{1}{s}\text{Var}(\nabla_\theta \ell(\mathbf{h}(\mathbf{x}; \theta), \mathbf{y}))$ where the last equality follows from the property of the variance of the sum of i.i.d. random variables. The above equation indicates that the inner gradient variance inversely scales with the number of shots, but the model is unaware of this fact, since it is only updated via the average gradient. Therefore, the model has no way to know that it should "trust" the gradient more when more shots are available, leading to poor performance in the variable shot setting. This is also shown empirically in Section 7.1.

### 4.2 THE LEARNING RATE SCALING METHOD

Intuitively, if we have a lower variance gradient, we can take larger steps along that gradient. Therefore, a natural way to vary the model update based on the number of datapoints is to change the inner learning rate. It is well known in standard supervised learning that the optimal learning rate scales with batch size (Smith & Le, 2018). One way to incorporate this observation into a variable-shot MAML algorithm would be to meta-learn *different* learning rates for each number of shots. As we will see in Section 7.1, this approach, which we call MAML-VL (variable-shot and learning), can indeed improve over standard MAML. However, this requires meta-learning an additional parameter for every possible number of shots, which can become impractical as the maximum number of shots increases.

Instead, we can derive a simple rule for the optimal learning, which we will show can work even *better* than meta-learning separate rates for each number of shots. First, let us define $\mathbf{U}_t(\theta, \alpha) = \theta - \alpha \nabla_\theta f_t(\theta)$ to be the ideal update rule given the average gradient. We can view this as the update rule for the limiting case of infinite shots, since $\mathbf{U}_t(\theta, \alpha) = \lim_{s \to \infty} \mathbf{U}_t(\theta, \alpha, s)$ in distribution by the law of large numbers. Let $\beta^*$ be the optimal learning rate such that $\mathbf{U}_t(\theta, \beta^*)$ attains optimal performance, i.e. $\beta^* = \arg \min_\beta \mathbb{E}_{\mathcal{T}_t} [f_t(\mathbf{U}_t(\theta, \beta))]$. This can be interpreted as the optimal inner

learning rate when $s \to \infty$. We want to find an optimal $s$-shot learning rate $\alpha_s^*$ that imitates this behavior for a finite $s$. Specifically, we solve for the value of $\alpha_s^*$ that minimizes the mean square error between the adapted parameters: $\alpha_s^* = \arg\min_\alpha \mathbb{E}_{\mathcal{T}_t}\left[||\mathbf{U}_t(\theta, \alpha, s) - \mathbf{U}_t(\theta, \beta^*)||_2^2\right]$.

**Theorem 1.** *The optimal learning rate for the $s$-shot case can be calculated as* $\alpha_s^* = \left(1 - \frac{1}{1+\frac{C_2}{C_1}s}\right)\beta^*$ *where* $C_1 = \mathbb{E}_{\mathcal{T}_t}\left[Var(\nabla_\theta \ell(\mathbf{h}(\mathbf{x}; \theta), \mathbf{y}))\right]$ *and* $C_2 = \mathbb{E}_{\mathcal{T}_t}\left[||\mathbb{E}_{\mathcal{D}_t} \nabla_\theta \ell(\mathbf{h}(\mathbf{x}; \theta), \mathbf{y})||_2^2\right]$.

We provide the proof in Appendix B. The quantities $C_1$, $C_2$ and $\beta^*$ are all functions of the parameters $\theta$, and thus require re-estimation each time we update $\theta$ during meta-training. In practice, we denote $\beta := \beta^*$ as the learned learning rate and $\eta := \frac{C_2}{C_1}$ as the learned scaling factor, and treat them as parameters that are meta-learned jointly with $\theta$. Denote $\alpha_s(\beta, \eta) = \left(1 - \frac{1}{1+\eta s}\right)\beta$ as the learnable scaled learning rate with parameters $\beta$ and $\eta$. In contrast to MAML-VL, which must meta-learn a number of learning rates that scales with the maximum number of shots, here we only meta-learn two additional scalar-valued parameters. This leads to our proposed model-agnostic meta-learning with variable-shot and scaling (MAML-VS) algorithm, whose meta-training objective is given below:

$$\min_{\theta,\beta,\eta} \frac{1}{N} \sum_{i=1}^N \mathbb{E}_{s \sim \text{Unif}(0,M)}\left[f_i(\theta - \alpha_s(\beta, \eta)\nabla_\theta \hat{f}_i^s(\theta))\right]. \tag{1}$$

## 5 ONLINE INCREMENTAL META-LEARNING WITH VARIABLE-SHOT MAML

With MAML-VS defined in the section above, we are ready to present the algorithm for the online incremental meta-learning setting. In following two subsections, we will discuss an online incremental meta-learning algorithm capable of variable-shot generalizations (Section 5.1) and its practical instantiation (Section 5.2).

### 5.1 FOLLOW THE META LEADER WITH VARIABLE-SHOT AND SCALING

We extend FTML (Finn et al., 2019) to the online incremental meta-learning setting by enabling the meta-learner to handle variable-shot data with the scaling rule in MAML-VS. Specifically, in the online incremental meta-learning setting, the model parameter $\theta$ and the scaled learning rate $\alpha_s(\beta, \eta)$ defined in Section 4.2 are updated in the following way:

$$\theta_{t+1}, \beta, \eta = \arg\min_{\theta,\beta,\eta} \sum_{j=1}^t \sum_{s=0}^{S_j} f_j(\mathbf{U}_j(\theta, \alpha_s(\beta, \eta), s)). \tag{2}$$

We term our online incremental meta-learning algorithm FTML-VS. Intuitively, for the current round $t+1$, FTML-VS plays the best variable-shot meta-learner in hindsight after each round based on all the meta-learned models attained in previous rounds $j = 1 \ldots t$. In practice, we of course cannot calculate the exact loss $f_j$, and therefore must approximate it using stochastic gradient descent methods, which will be presented in the next subsection.

### 5.2 FTML-VS IN PRACTICE

Optimizing Equation 2 is not feasible in practice as it has no closed form solution. We only have access to a portion of $\mathcal{T}_t$ in round $t$, and thus the population risk $f_t$ is impossible to compute exactly, as discussed by Finn et al. (2019). We adopt online stochastic gradient descent algorithms to resolve these issues. At step $s$ in round $t$, we compute gradients of our model parameters $\theta_t$ and scale learning rate $\alpha_K(\beta, \eta)$ where $K$ is the number of shots sampled in a minibatch using the following procedure:

1. Draw a task $\mathcal{T}_j : j \sim \kappa(t)$ (or a minibatch of tasks) uniformly from the set of tasks the agent has seen so far $\{\mathcal{T}_i\}_{i=1}^t$, where $\kappa(t)$ is the uniform distribution on $\{1, \ldots, t\}$. Let $\hat{\mathcal{D}}_j = \hat{\mathcal{D}}_j(s)$ if $j = t$ and otherwise $\hat{\mathcal{D}}_j = \hat{\mathcal{D}}_j(S_j)$, which are all the data accumulated for task $\mathcal{T}_j$ in the previous round.
2. Sample $K \sim \nu(j)$, where $\nu(j)$ is uniformly distributed on $\{0, \ldots, M(j)\}$ with $M(j) = \min\{M, |\hat{\mathcal{D}}_j|\}$.
3. Sample a minibatch $\mathcal{D}_j^{\text{tr}} \subset \hat{\mathcal{D}}_t(s)$ of size $K$ to compute the inner gradient updates using the derived scaled learning rate $\alpha_K(\beta, \eta)$.

4. Sample another minibatch $\mathcal{D}_j^{\text{val}} \subset \hat{\mathcal{D}}_t(s)$ to compute the gradients $\mathbf{g}^{\theta_t}, \mathbf{g}^\beta, \mathbf{g}^\eta$ for updating the model parameter $\theta_t$ along with parameters of scaled learning rate, $\beta$ and $\eta$ respectively.

where we denote $J = \mathbb{E}_{j\sim\kappa(t),K\sim\nu(j)} \left[ \mathcal{L}(\mathcal{D}_j^{\text{val}}, \mathbf{U}_t(\theta_t, \alpha_K(\beta, \eta), K)) \right]$, $\mathbf{U}_t(\theta_t, \alpha, K) = \theta_t - \alpha_K(\beta, \eta)\nabla_{\theta_t}(\mathcal{L}(\mathcal{D}_t^{\text{tr}}, \theta_t))$ and $\mathbf{g}^\theta = \nabla_{\theta_t} J$, $\mathbf{g}^\beta = \nabla_\beta J$ and $\mathbf{g}^\eta = \nabla_\eta J$. Note that in the case of zero-shot adaptation, i.e. $K = 0$, $\mathbf{U}_t(\theta_t, \alpha_K(\beta, \eta), K)$ corresponds to the parameter at the previous round, i.e. $\theta_{t-1}$.

We outline the complete algorithm in Algorithm 1 in Appendix A with some implementation details in Appendix C. The algorithm learns $\theta$, $\beta$, and $\eta$. We also include the training procedure and the evaluation protocol in Appendix A.

## 6 RELATED WORK

Prior work in meta-learning has studied how to acquire learning rules (Schmidhuber, 1987; Bengio et al., 1992; Hochreiter et al., 2001), accelerate optimization (Andrychowicz et al., 2016), and acquire priors suitable for few-shot learning (Finn et al., 2017). Modern meta-learning algorithms can broadly be classified into three high-level approaches: black-box meta-learners that parameterize a learning procedure using a neural network (Santoro et al., 2016; Ravi & Larochelle, 2017; Munkhdalai & Yu, 2017; Duan et al., 2016; Wang et al., 2016; Mishra et al., 2017), non-parametric meta-learners (Koch, 2015; Vinyals et al., 2016; Snell et al., 2017), and optimization-based meta-learners that embed an optimization procedure into the meta-learner (Finn et al., 2017; Li et al., 2017b; Rusu et al., 2018; Zintgraf et al., 2018; Bertinetto et al., 2018; Rajeswaran et al., 2019; Lee et al., 2019; Yoon et al., 2018; Finn et al., 2018). We choose to build upon the latter class of optimization-based meta-learners, since they produce a well-formed optimization process that tends to be robust to out-of-distribution tasks, a useful characteristic in online meta-learning settings.

While some works have evaluated meta-learning algorithms for $K$-shot $N$-way classification for a breadth of $K$ and $N$ (Snell et al., 2017; Triantafillou et al., 2019; Allen et al., 2019b;a; Hsu et al., 2018; Cao et al., 2020), these works typically train different networks for each value of $N$ and $K$, with some specifically noting poor generalization to values of $K$ not used in training (Snell et al., 2017; Hsu et al., 2018). Unlike these works, we consider the setting where a single meta-learned model must handle multiple different values of $K$. This setting has been considered in black-box methods that receive datapoints and make predictions incrementally (Santoro et al., 2016; Woodward & Finn, 2017), but this approach is known to underperform the setting where the model is trained for a specific value of $K$ (Mishra et al., 2017). Recent works have proposed non-parametric meta-learners that work effectively with variable numbers of shots (Allen et al., 2019b; Cao et al., 2020). Non-parametric methods such as these prior works require at least one shot per class, whereas we consider a setting where there is even fewer datapoints available. This is possible in settings with non-mutually-exclusive tasks, such that the meta-learned model can provide a sufficiently strong prior about the class labels.

Our variable-shot meta-learning setting is specifically motivated by a problem setting that resembles online learning (Shalev-Shwartz, 2012) and continual learning (Thrun, 1998). It is well known that algorithms such as follow the leader (FTL) are computationally-expensive, and many works in online learning and continual learning focus on developing more computationally-efficient methods (Kirkpatrick et al., 2017; Rebuffi et al., 2017; Lopez-Paz et al., 2017; French, 1999; Chaudhry et al., 2018; Zenke et al., 2017). Following multiple prior methods (Finn et al., 2019; Tessler et al., 2016; Rolnick et al., 2019), we instead focus on settings where it is practical to maintain a replay buffer of data, and focus on the problem of effective forward transfer when presented data from a sequence of tasks. We do so by combining employing meta-learning in the online setting. Unlike prior online meta-learning works (Finn et al., 2019; Grant et al., 2019; Khodak et al., 2019; Zhuang et al., 2019), we develop an algorithm that is specifically tailored to leverage variable amounts of data, allowing it to outperform a prior state-of-the-art online meta-learning approach (Finn et al., 2019). Finally, our work considers a problem setting that is distinct from works that a study online or continual learning in the *inner loop* of meta-learning (Al-Shedivat et al., 2017; Nagabandi et al., 2018; Javed & White, 2019; Denevi et al., 2019; He et al., 2019; Harrison et al., 2019).

## 7 Experiments

Our experiments aim to address the following questions, in both offline and online problem settings. In the more standard offline meta-learning setting, we aim to answer: **(1)** How does our variable-shot meta-learning method compare to prior methods, when evaluated with variable shots in the offline setting? **(2)** Does our theoretically motivated learning rate rule match the performance of learning per-shot learning rates? We then integrate our variable-shot method into the online setting, to study: **(3)** Does variable-shot learning improve the regret in the online incremental meta-learning setting? **(4)** Does our online incremental meta-learning algorithm attain better cumulative regret than standard empirical risk minimization? Details regarding the architecture of the models and the training setup are presented in Appendix D.

While our method can accommodate any standard meta-learning problem, we intentionally focus our evaluation on meta-learning problems with *non-exclusive* tasks, since this setting is particularly relevant in the online setting, which we will discuss in the subsection below and since this setting known to be especially difficult (Yin et al., 2019). With non-exclusive tasks, it is technically possible for the model to learn to solve all tasks in zero shot, which provides for a fair comparison to non-meta-learning online methods, such as the "follow the learder" method that trains on all data seen so far (indicated as "TOE" in Section 7.2). However, in order to attain the best possible regret, an effective meta-trained model must still learn to adapt, so as to solve new tasks quickly *before* it has learned to solve all tasks in zero shot. Standard meta-learning methods often perform poorly in this regime, due to a memorization problem (Yin et al., 2019), where they learn to only solve the meta-training tasks in zero shot, failing to either solve or adapt to the new task. We use three non-mutually exclusive datasets for our offline and online meta-learning experiments respectively and one mutually-exclusive dataset for the offline setting only, which are listed as follows:

**(1) (Non-mutually exclusive) Rainbow MNIST (Finn et al., 2019).** Based on the MNIST digit recognition dataset, this dataset consists of digits transformed in different ways: 7 backgrounds with different colors, 2 scales (half and original size), and 4 rotations of 90 degree intervals, which leads to 56 total tasks in combination. Each task contains 900 images applied with one of the transformation listed above (See Appendix D for visualization); **(2) (Non-mutually exclusive) Contextual MiniImagenet.** This dataset, adapted from MiniImagenet (Ravi & Larochelle, 2016), is meant to provided a non-mutually-exclusive variant of the MiniImagenet few-shot recognition task, such that non-meta-learning baselines can feasibly solve this task. We use the same images and classes as MiniImagenet, but formulate each class as a separate binary classification problem. The model receives two images as input: a reference image that determines the current task, and a query image, and the goal is to output a binary label indicating whether the query image belongs to the same class as the reference image, or not. In this case, each class in the MiniImagenet constitutes a task, which makes the task space large and the problem challenging. We include a visualization of the architecture for solving this dataset in Figure 3 in Appendix D.3; **(3) (Non-mutually exclusive) Pose Prediction (Finn et al., 2019; Yin et al., 2019).** In this setting, we construct a multi-task dataset with 65 tasks based on the Pascal 3D data (Xiang et al., 2014) where each task corresponds to a different object with a random camera angle. We render the image each object using MuJoCo (Todorov et al., 2012) with a random orientation. We only use this dataset in the online setting; **(4) (Mutually exclusive) Omniglot (Lake et al., 2011).** This dataset has 20 instances of 1623 characters from 50 different alphabets where each instance is drawn by a different person. We only use this dataset in the offline setting and include the results in Appendix F.

### 7.1 Offline Meta-Learning Experiments

We first evaluate our variable-shot methods, MAML-VS and MAML-VL, in the offline setting, in comparison to MAML (Finn et al., 2017) and MANN (Santoro et al., 2016), which uses a non-gradient based recurrent meta-learner. The offline experiments mainly serve as a sanity check of our online incremental experiments, where variable-shot learning is practically important. Experiment setup details, including descriptions of baselines, the dataset setup, and model architectures can be found in Appendix D. Note that as discussed in Section 6, non-parametric meta-learning methods require at least one shot for each class, making them inapplicable to non-mutually exclusive tasks where the total number of support datapoints is often much less than the total number of classes. Hence, we choose not to compare to those approaches. As shown in Table 1, our method MAML-VL achieves the best performance in 5 out of 8 total settings in both datasets while MAML-VS achieves the best performance in the 20-shot classification accuracy on the Contextual MiniImagenet, which

| Dataset | Method | 0-Shot | 1-Shot | 10-Shot | 20-Shot |
|---|---|---|---|---|---|
| Rainbow MNIST | MANN (Santoro et al., 2016) | $66.41 \pm 1.55$ | $66.49 \pm 1.68$ | $67.09 \pm 1.38$ | $66.95 \pm 1.12$ |
| | MAML (Finn et al., 2017) | $71.30 \pm 0.53$ | $71.13 \pm 0.91$ | $74.78 \pm 1.12$ | $\mathbf{78.67} \pm 0.68$ |
| | MAML-VL (ours) | $\mathbf{73.14} \pm 0.65$ | $\mathbf{73.45} \pm 0.61$ | $\mathbf{74.87} \pm 0.38$ | $76.59 \pm 0.36$ |
| | MAML-VS (ours) | $72.72 \pm 0.45$ | $72.91 \pm 0.15$ | $73.89 \pm 0.60$ | $77.17 \pm 0.10$ |
| Contextual MiniImagenet | MANN (Santoro et al., 2016) | $\mathbf{64.85} \pm 0.04$ | $\mathbf{64.90} \pm 0.19$ | $64.78 \pm 0.63$ | $65.50 \pm 0.50$ |
| | MAML (Finn et al., 2017) | $61.36 \pm 0.48$ | $56.80 \pm 0.65$ | $63.82 \pm 0.09$ | $63.84 \pm 0.31$ |
| | MAML-VL (ours) | $61.17 \pm 1.95$ | $59.11 \pm 3.06$ | $\mathbf{66.09} \pm 2.93$ | $66.15 \pm 4.74$ |
| | MAML-VS (ours) | $63.37 \pm 0.71$ | $63.19 \pm 1.14$ | $65.89 \pm 0.10$ | $\mathbf{66.94} \pm 0.14$ |

Table 1: **Complete Results for Offline Rainbow MNIST and Offline Contextual MiniImagenet.** Comparison of MAML, MANN, MAML-VL (ours) and MAML-VS (ours). The results are classification accuracies averaged over 3 random seeds, $\pm$ one standard deviation. On Rainbow MNIST, MAML-VL (ours), that learns a learning rate per shot, does better in 0, 1, and 10-shot setting, which is the most effective in this relatively simple digit classification task. MAML-VS (ours) achieves comparable performance in all four settings in this experiment. On Contextual MiniImagenet, which is a challenging image classification problem with large task space, MAML-VS (ours) achieves better performance in 20-shot classification accuracies while maintaining competitive performances in 0, 1 and 10-shot settings. Overall, MAML-VS achieves the best or comparable performances in all 8 settings, which indicates that our theoretically-motivated inner learning rate scaling rule would excel in the online setting, which is empirically shown in Table 2.

| Method | Incremental Rainbow MNIST | Incremental Contextual MiniImagenet | Incremental Pose Prediction |
|---|---|---|---|
| TOE (Finn et al., 2019) | $16516.6 \pm 172.5$ | $2314.5 \pm 363$ | $125.2 \pm 0.6$ |
| FTML (Finn et al., 2019) | $4804.2 \pm 302.8$ | $1037.7 \pm 116.1$ | $135.5 \pm 18.8$ |
| FTML+Meta-SGD (Li et al., 2017a) | $4699.0 \pm 212.0$ | $1027.3 \pm 35.8$ | $125.0 \pm 9.4$ |
| FTML-VL (ours) | $4502.7 \pm 477.0$ | $1033.9 \pm 21.5$ | $141.3 \pm 30.3$ |
| FTML-VS (ours) | $\mathbf{4484.7} \pm 133.8$ | $\mathbf{1020.0} \pm 40.8$ | $\mathbf{119.1} \pm 3.2$ |

Table 2: Final cumulative regret on Incremental Rainbow MNIST, Incremental Contextual MiniImagenet and Incremental Pose Prediction. Results are cumulative regrets averaged over 3 random seeds, $\pm$ one standard deviation. FTML-VS outperforms other methods in all three domains.

is a challenging problem with large task space and attains comparable performance in all other 7 settings. In both experiments, training for variable shot with a single learning rate (MAML) causes the algorithm to sacrifice 0-shot and 1-shot performance in favor of better performance with more shots (10 and 20). In the Contextual MiniImagenet setting especially, 1-shot performance is *significantly* lower than even the 0-shot performance. For MANN, variable shot training instead biases the model toward good 0-shot and 1-shot performance, and sacrifices performance with more shots. We expect both issues to be problematic in the online setting: good few-shot performance is important in the later stages of training, where each task can be mastered with just a few samples, and good many-shot performance is critical for difficult tasks or early on in training. Our method, MAML-VS and MAML-VL, performs well with all numbers of shots, and performance improves as more examples are observed.

## 7.2 ONLINE INCREMENTAL META-LEARNING EXPERIMENTS

To answer questions **(3)** and **(4)**, we evaluate FTML-VS in the online incremental setting. To evaluate the performance of our approach, we compare our FTML-VS to: (1) **Train on everything (TOE) (Finn et al., 2019)**, which is the standard empirical risk minimization approach that trains a single predictive model on all the data available so far, i.e. $\{\hat{D}_i\}_{i=1}^t$. The trained model is directly tested on $\mathcal{D}_t^{\text{test}}$ without any adaptation; (2) **FTML (Finn et al., 2019)**, which trains an online meta-learner with fixed inner learning rate and is evaluated on the adaptation performance on $\mathcal{D}_t^{\text{test}}$; (3) **FTML-VL (Ours)**, which trains an online meta-learner with a learned inner learning rate for each number of shots ranging from 0 to $M$; (4)FTML+Meta-SGD (Li et al., 2017a), which learns per-parameter learning rates for FTML; (5) **Incremental ProtoNet (Snell et al., 2017)**, which adapts Snell et al. (2017), one of the most widely used non-parametric meta-learning method to the online incremental meta-learning setting; (6)**A-GEM (Chaudhry et al., 2018)**, which is a popular continual learning algorithm that prevents catastropic forgetting.

For **Incremental Rainbow MNIST**, we set the proficiency threshold $C$ at $85\%$ classification accuracy computed on a minibatch of test data $\mathcal{D}_t^{\text{test}} \subset \mathcal{D}_t$ where $\mathcal{D}_t^{\text{test}} \cap \hat{D}_t(s) = \emptyset$ for each task $t$, and move on to the next task when this threshold is crossed or after 2000 steps. For **Incremental Contextual MiniImagenet**, we set the proficiency threshold $C$ at $75\%$ classification accuracy computed on $\mathcal{D}_t^{\text{test}}$.

| Dataset | A-GEM (Chaudhry et al., 2018) | FTML-VS (ours) |
|---|---|---|
| Incremental Rainbow MNIST | $14292.19 \pm 201.72$ | $4484.70 \pm 113.83$ |

Table 3: Comparison between A-GEM (Chaudhry et al., 2018) and FTML-VS on Incremental Rainbow MNIST. Results are cumulative regrets averaged over 3 random seeds, $\pm$ one standard deviation. FTML-VS achieves superior results compared to A-GEM.

| Dataset | Incremental ProtoNet (Snell et al., 2017) | FTML-VS (ours) |
|---|---|---|
| Incremental Rainbow MNIST | $34812.7 \pm 6197.7$ | $19207.7 \pm 282.4$ |

Table 4: Comparison between Incremental ProtoNet (Snell et al., 2017) and FTML-VS on Incremental Rainbow MNIST with adapted data-generating scheme discussed in Section 7.2. Results are cumulative regrets averaged over 3 random seeds, $\pm$ one standard deviation. FTML-VS significantly outperforms Incremental ProtoNet.

For **Incremental Pose Prediction**, we consider mean-square error as the metric and do not advance the task automatically. We discuss the details of the online procedure of each dataset in Appendix E. We evaluate the performance based on the final cumulative regret $\text{Regret}_T$, which is computed by accumulating the test losses on $\mathcal{D}_t^{\text{test}}$ during evaluation time for each task $t$, as shown in Table 2 and include the regret curves over the online incremental meta-learning process of all methods in all domains in Appendix E. We also include additional ablation studies in Appendix G.

According to Table 2, in all three domains, FTML-VS attains smallest regret after training all the tasks sequentially compared to other methods, suggesting the importance of using our learning rate rule in the online incremental meta-learning setting with non-mutually exclusive datasets. Moreover, FTML-VS outperforms TOE by a large margin, which indicates that our online meta-learning method is able to solve tasks more efficiently than empirical risk minimization by leveraging the shared task structure to quickly adapt to each new task. Finally, FTML-VS also achieves lower final cumulative regret than FTML + Meta-SGD, suggesting that our theoretically sound rule of selecting learning rates for different number shots is more effective than simply learning learning rates for each parameter. It is also worth noting that learning per-parameter learning rates is complementary to our method and can be combined to further improve performances. We will leave this for future work.

Moreover, we conduct the comparison between FTML-VS and A-GEM on the Incremental Rainbow MNIST dataset with the same online procedure described above. As shown in Table 3, FTML-VS achieves much better cumulative regret compared to A-GEM. This result is unsurprising, as prior continual learning works focus primarily on minimizing negative backward transfer and compute considerations, as opposed to our goal of accelerating forward transfer through meta-learning.

Finally, for the comparison to Incremental ProtoNet, we consider the Incremental Rainbow Mnist dataset. As described in Section 7.1, non-parametric meta-learning methods is applicable to settings where the number of shots is smaller than the number of classes. Hence, in order to adapt Snell et al. (2017), when we do not have the support datapoints for a particular class, i.e. a digit, of the current task, we compute the prototype of this class by sampling data of this class from previous tasks. We compare FTML-VS to Incremental ProtoNet following such a data-generating protocol and the same online procedure as in the Incremental Rainbow MNIST experiment described in the paragraph above. As shown in Table 4, FTML-VS outperforms Incremental ProtoNet by a significant margin, suggesting that our theoretically motivated learning rate selection rule is pivotal in online incremental meta-learning setting compared to non-parametric meta-learning methods, which require the minimum number of shots to be greater or equal to the number of classes.

## 8    CONCLUSION

In this work, we studied meta-learning in the context of a sequence of tasks. While most meta-learning works study how to achieve few-shot adaptation, succeeding in sequential learning settings requires both meta-training and adaptation to be efficient and performant. Unlike prior works, we focused on an online meta-learning setting where data for each new task arrives incrementally. Motivated by the challenges of this setting, we introduced a variable-shot meta-learning algorithm that optimizes for good performance after adapting with varying amounts of data. Our approach introduces a scaling rule for the learning rate that scales with the number of shots. This approach strongly outperformed a strictly more expressive approach of learning individual learning rates for each number of shots, validating the correctness of our derivation. On both offline and online meta-learning settings, we observe significant benefits from our approach compared to prior methods.

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

# A    ALGORITHM OUTLINE

---

**Algorithm 1** Online Incremental Meta-Learning with FTML-VS

---

1: **Input:** Performance threshold of proficiency, $C$
2: randomly initialize $\theta_1, \theta_1', \beta, \eta$
3: initialize the task buffer as empty, $\mathcal{B} \leftarrow [\,]$
4: initialize the regret $\text{Regret}_T \leftarrow 0$
5: **for** $t = 1, \ldots$ **do**
6:     initialize $\hat{\mathcal{D}}_t(0) = \emptyset$
7:     Add $\mathcal{B} \leftarrow \mathcal{B} + [\,\mathcal{T}_t\,]$
8:     initialize $s \leftarrow 0$
9:     **while** $\mathcal{L}\left(\mathcal{D}_t^{\text{test}}, \theta_t'\right) > C$ **do**
10:         Add datapoints $\{(\mathbf{x}, \mathbf{y})\}$ to $\hat{\mathcal{D}}_t(s)$ until $|\hat{\mathcal{D}}_t(s)| = g(s)$
11:         $\theta_t, \alpha, \eta \leftarrow \texttt{VS-Meta-Update}(\theta_t, \beta, \eta, \mathcal{B}, t, s)$
12:         **if** $|\hat{\mathcal{D}}_t(s)| > 0$ **then**
13:             $\theta_t' \leftarrow \texttt{VS-Update-Procedure}\left(\theta_t, \beta, \eta, \hat{\mathcal{D}}_t(s)\right)$
14:         **else**
15:             $\theta_t' \leftarrow \theta_t$
16:         **end if**
17:         $s \leftarrow s + 1$
18:         $\text{Regret}_T \leftarrow \text{Regret}_T + \mathcal{L}\left(\mathcal{D}_t^{\text{test}}, \theta_t'\right)$
19:     **end while**
20:     Record regret up until task $\mathcal{T}_t$ using $\text{Regret}_T$
21:     Set $S_t \leftarrow s$
22:     $\theta_{t+1} \leftarrow \theta_t$
23: **end for**

---

**Algorithm 2** FTML-VS Subroutines

---

1: **Input:** Hyperparameters parameters $\gamma, M$
2: **function** $\texttt{VS-Meta-Update}(\theta, \beta, \eta, \mathcal{B}, t, s)$
3:     **for** $n_{\text{m}} = 1, \ldots, N_{\text{meta}}$ steps **do**
4:         Sample task $\mathcal{T}_j$: $j \sim \kappa(t)$ // (or a minibatch of tasks)
5:         Sample the random number of shot $K \sim \nu(j)$.
6:         Sample minibatches $\mathcal{D}_j^{\text{tr}}$ with $K$ datapoints and $\mathcal{D}_j^{\text{val}}$ uniformly from $\hat{\mathcal{D}}_t(s)$
7:         Compute gradient $\mathbf{g}_t^\theta, \mathbf{g}_t^{\alpha_K}$ using $\mathcal{D}_j^{\text{tr}}, \mathcal{D}_j^{\text{val}}$, and Procedure 4
8:         Update parameters $\theta \leftarrow \theta - \gamma\, \mathbf{g}_t^\theta$   // (or use Adam)
9:         Update $\beta \leftarrow \alpha - \gamma\mathbf{g}_t^{\alpha_K} \cdot \nabla_\beta \alpha_K$ and $\eta \leftarrow \eta - \gamma\mathbf{g}_t^{\alpha_K} \cdot \nabla_\eta \alpha_K$
10:     **end for**
11:     Return $\theta, \alpha, \eta$
12: **end function**
13: **function** $\texttt{VS-Update-Procedure}(\theta, \beta, \eta, \mathcal{D})$
14:     **if** $|\mathcal{D}| > M$ **then**
15:         Sample a minibatch $\hat{\mathcal{D}} \sim \mathcal{D}$ with $|\hat{\mathcal{D}}| = M$ and set $\mathcal{D} \leftarrow \hat{\mathcal{D}}$
16:     **end if**
17:     **for** $n_{\text{g}} = 1, \ldots, N_{\text{grad}}$ steps **do**
18:         $\theta \leftarrow \theta - \beta \cdot \left(1 - \frac{1}{1+\eta|\mathcal{D}|}\right) \nabla \mathcal{L}(\mathcal{D}, \theta)$
19:     **end for**
20:     Return $\theta$
21: **end function**

---

For each task $t$, we discuss the training procedure and evaluation protocol below.

*At training time*, we maintain a dataset $\hat{D}_t(s)$ that receives one datapoint incrementally at each step $s$ of task $\mathcal{T}_t$, i.e. $|\hat{D}_t(s)| = s$. After appending data to $\hat{D}_t(s)$, the model parameters $\theta$, learned learning rate $\beta$, and the learned scaling factor $\eta$ are updated as per Procedure 4 following the procedure discussed above and in $\texttt{VS-Meta-Update}$ in Algorithm 2 in Appendix A.

*During evaluation*, we sample a minibatch of test data $\mathcal{D}_t^{\text{test}} \subset \mathcal{D}_t$ where $\mathcal{D}_t^{\text{test}} \cap \hat{D}_t(s) = \emptyset$. We compute the adapted parameter $\theta_t'$ using $\texttt{VS-Meta-Update}$, using either all the data for task $t$ so far, or a maximum of $M$ datapoints if more than $M$ datapoints have been collected. Then, we

compute the loss on the test set $\mathcal{D}_t^{\text{test}}$ using $\theta'$. Hence, the online incremental meta-learner is evaluated based on the performance of adaptation to unseen data with an incremental number of shots ranging from $0$ to $M$. If the loss on the test set reaches the proficiency threshold $C$ (e.g., $90\%$ accuracy), we advance to the next task and record the data efficiency of $\mathcal{T}_t$.

The optimal scaled learning rate for each number of shots is computed as $\alpha_K(\beta, \eta) = \beta \left(1 - \frac{1}{1+\eta K}\right)$. We keep a buffer $\mathcal{B}$, initialized as empty, to store the task data seen so far. The cumulative regret $\text{Regret}_T$ is computed by accumulating the test losses during evaluation time for each task $t$. If the model learns to advance to the next task with improving data efficiency, the cumulative regret would tend to level off over number of tasks, which suggests that we achieve adaptation from many-shot to few-shot over the course of the online incremental meta-learning.

## B    PROOF OF THEOREM 1

*Proof.* We apply variance and bias decomposition for the mean squared error to obtain:

$$\mathbb{E}_{\mathcal{T}_t}\left[||\mathbf{U}_t(\theta, \alpha, s) - \mathbf{U}_t(\theta, \beta^*)||_2^2\right]$$
$$=\mathbb{E}_{\mathcal{T}_t}\left[\mathbb{E}_{\mathcal{D}_t}\left[||\mathbf{U}_t(\theta, \alpha, s) - \mathbb{E}_{\mathcal{D}_t}\mathbf{U}_t(\theta, \alpha, s)||_2^2\right]\right] + \mathbb{E}_{\mathcal{T}_t}\left[||\mathbb{E}_{\mathcal{D}_t}\mathbf{U}_t(\theta, \alpha, s) - \mathbf{U}_t(\theta, \beta^*)||_2^2\right]$$
$$=\alpha^2 \frac{1}{K}\mathbb{E}_{\mathcal{T}_t}\left[\text{Var}(\nabla_\theta \ell(\mathbf{h}(\mathbf{x};\theta), \mathbf{y}))\right] + (\alpha - \beta^*)^2 \mathbb{E}_{\mathcal{T}_t}\left[||\mathbb{E}_{\mathcal{D}_t}\nabla_\theta \ell(\mathbf{h}(\mathbf{x};\theta), \mathbf{y})||_2^2\right]$$

Minimizing this quadratic function with respect to $\alpha$, we get

$$\alpha_s^* = \left(1 - \frac{1}{1 + \frac{C_2}{C_1}s}\right)\beta^*$$

with the $C_1$ and $C_2$ as defined in Theorem 1. $\qquad\square$

## C    IMPLEMENTATION DETAILS FOR FTML-VS

In practice, we find that using multiple inner gradient updates in $\mathbf{U}_t(\theta, \alpha_K(\beta, \eta), K)$ improves the performance of the method.

Moreover, it is also helpful to add meta-regularization (Yin et al., 2019) to improve the performance on few-shot adaptation in our variable-shot learning and online incremental meta-learning settings. Specifically, we model the parameters with stochasticity and at each round $t$, $\theta_t \sim q(\theta; \theta_\mu, \theta_\sigma)$ is drawn from a distribution $q$ with mean $\theta_\mu$ and variance $\theta_\sigma$. We typically use $q(\theta; \theta_\mu, \theta_\sigma)$ as a normal distribution. Following Yin et al. (2019), we regularize the meta-objective by adding a $D_{\text{KL}}(q(\theta; \theta_\mu, \theta_\sigma)||r(\theta))$ term to Procedure 4, where $r(\theta)$ is a prior distribution on the model parameters (e.g., a standard normal distribution).

## D    DETAILED INFORMATION FOR OFFLINE EXPERIMENT SETUP

### D.1    VISUALIZATION OF DATASETS

We provide visualizations of images in Rainbow MNIST and Pose Prediction in Figure 2. For visualizations of Contextual MiniImagenet, see Figure 3.

### D.2    RAINBOW MNIST

#### D.2.1    DATASET

We use the same dataset as the Incremental Rainbow MNIST dataset in (Finn et al., 2019). Due to the relatively low zero-shot difficulty of the dataset, we only take the first 20 tasks as the training set.

#### D.2.2    MODEL ARCHITECTURES

We use the same neural network architecture for MAML and MAML-VS. The network consists of 4 3x3 kernel 32 channel convolution layers with leaky ReLU activation, and a final fully connected layer to output logits for 10 classes. For MANN, we first have an embedding layer that converts the 10-class adaptation labels to 256 dimensional vectors. We pass the adaptation image through a feature network consists of 4 3x3 kernel 32 channel convolution layers, followed by a fully connected

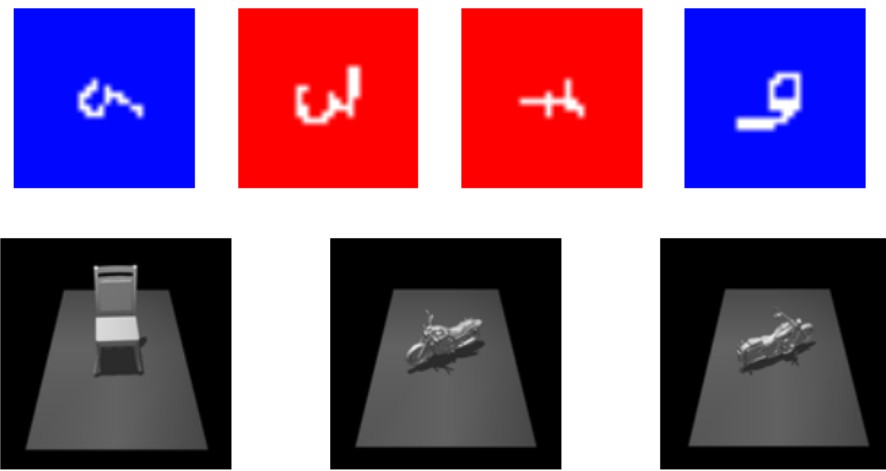

Figure 2: Visualizations of Rainbow MNIST (**Top**) and Pose Prediction (**Bottom**).

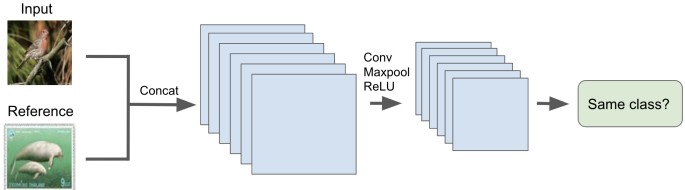

Figure 3: We visualize the Siamese network architecture for Contextual MiniImagenet, which takes in a pair of input and reference images and classify if they belong to the same class.

layer that output 256 dimensional image features. We then concatenate these image features with the label embeddings, and feed it into a GRU layer. The GRU layer outputs a 256 dimensional summary for the adaptation data. We then pass the test images into the same image feature network to obtain a 256 dimensional test image features. We concatentate the GRU layer's output with the test image features, and feed them into a final fully connected layer to obtain the 10 logits.

### D.3    CONTEXTUAL MINIIMAGENET

We present a visualization of the setup in Figure 3.

### D.3.1    DATASET

We use the same dataset as the MiniImageNet dataset in (Finn et al., 2017; Snell et al., 2017). In order to make the tasks non-mutually-exclusive, we cast it into a binary classification problem where the model receives an ordered pair of two images and classifies whether they belong to the same class. Different tasks in this dataset then correspond to the category of the first image in the pair. Each time we sample a batch, we sample half batch of pairs with images from the same class and half batch of pairs with images from different tasks. We use the first 90 classes as our training set.

### D.3.2    MODEL ARCHITECTURES

We use the same neural network architecture for MAML and MAML-VS. The network consists of 2 3x3 kernel 64 channel convolution layers with leaky ReLU activation, followed by a 2x2 max pooling layer, followed again by 2 3x3 kernel 64 channel convolution layers with leaky ReLU activation, followed by a 2x2 max pooling layer and a final fully connected layer that ouputs 2 logits. For MANN, the model is the same as we used in Rainbow MNIST experiments, except we use a different image feature network consists of 2 3x3 kernel 64 channel convolution layers with leaky ReLU activation, followed by a 2x2 max pooling layer, followed again by 2 3x3 kernel 64 channel convolution layers with leaky ReLU activation, followed by a 2x2 max pooling layer and a final fully connected layer that ouputs 256 dimensional image embeddings.

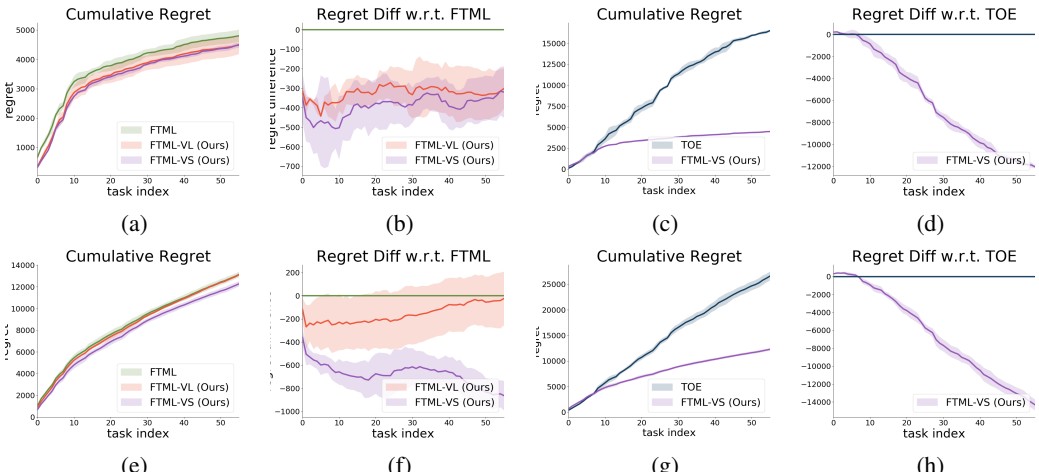

Figure 4: Curves of regret versus number of tasks on Incremental Rainbow MNIST with 3 random seeds. **Top row**: with automatic advancement to the next task based on proficiency on the current task. **Bottom row**: training each task with fixed number of steps.

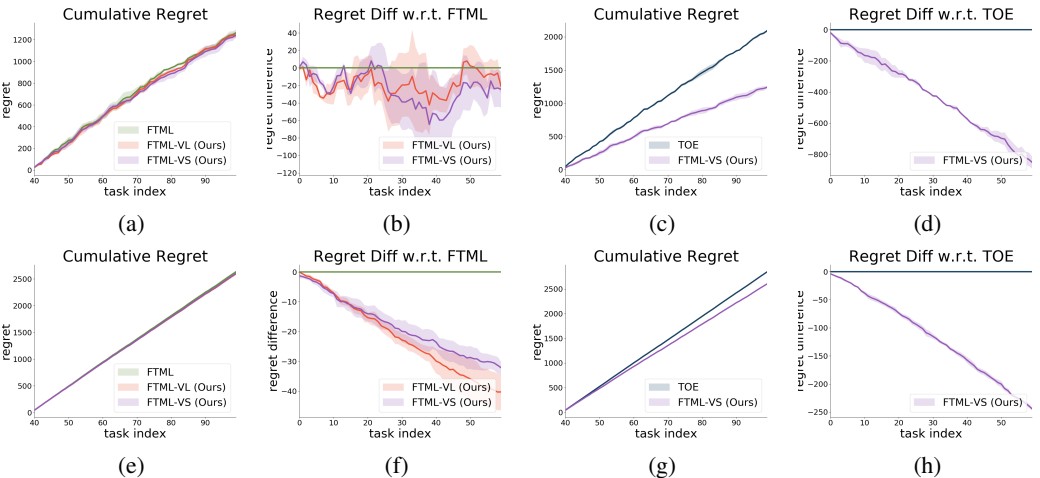

Figure 5: Curves of regret versus number of tasks on Incremental Contextual MiniImagenet with 3 random seeds. **Top row**: with automatic advancement to the next task based on proficiency on the current task. **Bottom row**: training each task with fixed number of steps.

### D.4 TRAINING FOR VARIABLE NUMBER OF SHOTS

During training time, we train the model with variable number of shots by averaging the loss for different shots. For zero-shot, we directly use the prior parameter without adaptation.

## E DETAILED INFORMATION FOR ONLINE INCREMENTAL EXPERIMENT SETUP

### E.1 INCREMENTAL RAINBOW MNIST

During training, we randomly sample a permutation of the 56 classes in Rainbow MNIST, where each class constitutes a task. For each task $t = 1, \ldots, 56$, we add 4 datapoints to $\hat{D}_t(s)$ every 10 steps. We set the maximum number of shots to be evaluated at $M = 20$. When $|\hat{D}_t(s)| \in \{0, 4, 8, \ldots, 40\}$, we evaluate $\frac{|\hat{D}_t(s)|}{2}$-shot performance, using one half of the task dataset for adaptation (i.e., $\mathcal{D}_t^{\text{train}}$) and the other half for evaluation (i.e., $\mathcal{D}_t^{\text{val}}$). Note that TOE does not perform few-shot adaptation, but *does* use the available data for the new task for standard supervised training.

For meta-training, we sample 25 tasks from the buffer $\mathcal{B}$ at each step, and randomly sample the number of shots $K \sim \text{Unif}\{0, 1, 2, \ldots, 20\}$. We use the same architecture of the model as in the offline setting. To train FTML, FTML-VL, FTML-VS, we take 5 inner gradient steps with inner gradient initialized at $0.1$. Inner gradient magnitudes are clipped within $10$. We train all methods

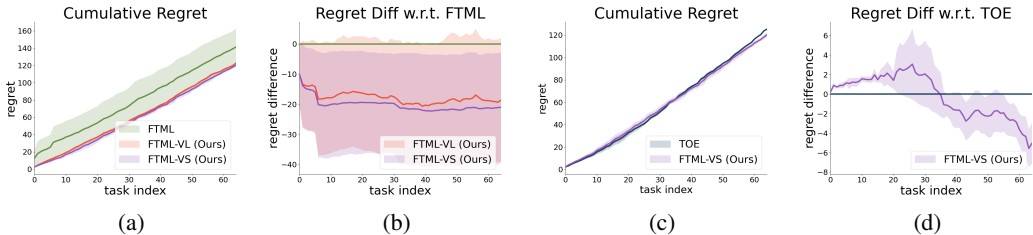

Figure 6: Curves of regret versus number of tasks on Incremental Pose Prediction with 3 random seeds.

using Adam with learning rate $0.0001$ with gradient clipping $10$. As discussed in Section 5.2, we use meta-regularization (Yin et al., 2019) with coefficient $0.1$. For FTML-VS, we initialize $\eta = 1.0$.

We run all methods with 3 random seeds with and without the automatic advancement to the next task based on proficiency on the current task discussed in Section 5.2 and include the results in Figure 4. We add 2 datapoints to $\hat{\mathcal{D}}_t(s)$ every 5 and 2 steps for with and without automatic advancement experiments respectively. We train maximum 2000 steps for each task in the setting where automatic advancement happens. In this setting, we advance the next task once the model achieves over $85\%$ success rate on the current task. In the case where we don't advance the task based on the current task performance, we train each task for fixed 800 steps. As shown in Figure 4, FTML-VS outperforms the baselines in both experimental settings, where the advantage of FTML-VS over other baselines is more apparent in the setting where automatic advancement doesn't occur.

### E.2 INCREMENTAL CONTEXTUAL MINIIMAGENET

As before, we generate a randomly shuffled sequence of 100 classes from MiniImagenet. We pretrain all methods on the first 40 tasks and then feed the next 60 tasks incrementally. We add 8 datapoints to the dataset of the current task every 20 steps, for a maximum of 1400 steps per task. We set the maximum number of shots to be $M = 20$ as before. We set the proficiency threshold $C$ at $75\%$ classification accuracy. When $|\hat{D}_t(s)| \in \{0, 8, 16, \ldots, 80\}$, we evaluate $\frac{|\hat{D}_t(s)|}{4}$-shot performance of all the methods except TOE, for the same reason as before.

During training, we sample 16 tasks from $\mathcal{B}$ and randomly sample $K \sim \text{Unif}\{0, 2, 4, \ldots, 20\}$. We also use the same architecture of the model as in the offline setting. Similar to Incremental Rainbow MNIST, we take 5 inner gradient steps with inner gradient initialized at $0.03$. We train all methods using Adam with learning rate $0.0001$ with gradient clipping $10$. We also use meta-regularization (Yin et al., 2019) with coefficient $0.1$. For FTML-VS, we initialize $\eta = 1.0$.

Similar to evalutions of Incremental Rainbow MNIST, We also include the results of 3 random seeds on settings with and without automatic advancement in Figure 5. We add 8 datapoints to $\hat{\mathcal{D}}_t(s)$ every 20 and 10 steps for with and without automatic advancement experiments respectively. For automatic advancement, we train each method with maximum 1400 steps and pick the proficiency threshold at $75\%$. In the setting without automatic advancement, we train each task for fixed 700 steps. As shown in Figure 5e-5h, FTML-VS outperforms the FTML but gets slightly higher overall regret compared to FTML-VL where automatic advancement doesn't happen whereas in Figure 5a-5d, FTML-VS achieves the minimal regret among all methods over the course of training.

### E.3 INCREMENTAL POSE PREDICTION

Similarly, we construct a random sequence of the 65 tasks in the 3D pose dataset. We add 4 datapoints to the dataset of the current task every 10 steps, for a maximum of 240 steps per task. We set the maximum number of shots to be $M = 15$. When $|\hat{D}_t(s)| \in \{0, 4, 8, \ldots, 96\}$, we evaluate $\frac{|\hat{D}_t(s)|}{2}$-shot performance as in Incremental Rainbow MNIST. In this setting, we consider mean-square error and does not advance the task automatically. We set the proficiency threshold $C$ at $85\%$ classification accuracy, and move on to the next task when this threshold is crossed or after 2000 steps.

As before, we present the cumulative regret curves of 3 random seeds on the Incremental Pose Prediction dataset. As shown in Figure 6b -6d, FTML-VS attains the best regret among all approaches. Note that TOE outperforms vanilla FTML in this dataset despite that FTML-VS achieves better cumulative regret over TOE, suggesting that empirical risk minimization works better than vanilla online meta-learning in this case while our method can further improve the performance of vanilla online meta-learning approaches and surpass the level of empirical risk minimization.

## F  OFFLINE EXPERIMENTS ON MUTUALLY EXCLUSIVE TASKS

We compare out method against MAML in the variable-shot settings on the Omniglot dataset, which has mutually exclusive tasks. We test 50-way classification on 1, 2 and 5 shots in order to make the task more difficult. The results are shown in Table 5. We can see that our method performs about the same as our baseline method MAML. This is because in the mutually exclusive settings, the number of datapoints of $K$-shot classification for computing inner gradient adaptation is $K$ times the number of classes, compared to merely $K$ in the mutually non-exclusive setting. For example, in this case, 5-shot corresponds to 250 images for computing inner gradient updates. This is a fairly large number of images so that the gradient might have little variance, and therefore a large gradient step could be taken without overfitting.

| Method | 1-Shot | 2-Shot | 5-Shot |
|---|---|---|---|
| MAML | $90.94 \pm 0.44$ | $95.29 \pm 0.19$ | $96.81 \pm 0.07$ |
| MAML-VL (ours) | $91.08 \pm 0.28$ | $95.35 \pm 0.26$ | $96.84 \pm 0.10$ |
| MAML-VS (ours) | $90.94 \pm 0.12$ | $95.05 \pm 0.23$ | $96.66 \pm 0.20$ |

Table 5: Offline 50-way Omniglot results.

## G  ABLATION STUDIES OF ONLINE INCREMENTAL EXPERIMENTS

| Method | $M = 10$ | $M = 20$ | $M = 30$ |
|---|---|---|---|
| FTML | $7710.0 \pm 769.8$ | $4804.2 \pm 302.8$ | $4250.7 \pm 253.6$ |
| FTML-VS (ours) | $5643.3 \pm 149.0$ | $4484.7 \pm 133.8$ | $3969.0 \pm 203.7$ |

Table 6: Ablation study of $M$, the maximum number of shots, on Incremental Rainbow MNIST.

We conduct an ablation study on the sensitivity of $M$, the maximum number of shots, on the incremental Rainbow MNIST dataset by comparing FTML and FTML-VS. As shown in Table 6, as M increases, the cumulative regret of both FTML and FTML-VS decreases since learning becomes easier with larger numbers of shots. Meanwhile, FTML-VS attains better performance compared to FTML with different values of M and the performance gap becomes larger as M decreases, suggesting that our theoretically motivated learning rate scaling rule based on the number of shots is important for different values of the maximum number of shots and is particularly effective when the maximum number of shots is small.

