# OpenReview forum: "Variable-Shot Adaptation for Online Meta-Learning"
_ICLR.cc/2021/Conference — Reject_

### Official Review · AnonReviewer4 · 2020-10-27
**The "new" task is not well-motivated; A more reasonable baseline could already tackled the "new" task; Meta-learning LR is not novel.**

**Rating:** 5
**Confidence:** 4

**Review:**

This paper defined a new problem called “variable-shot adaptation for incremental meta-learning”. In the proposed problem, the data within each task arrives one data point at a time, and the goal of the model is to minimize the cumulative regret summed over all of the tasks. It also proposed an algorithm that aimed to address the problem by using a scaling rule for the learning rate that scales with the number of shots. This algorithm is evaluated on four datasets.

Pros:

1- The idea of using a scaling rule for the learning rate that scales with the number of shots is interesting. The authors also provided the proof in the appendix.

2- The method is technically sound, and the experiment results can show its efficiency in some of the settings.

Cons:

1- About the problem formulation. This paper defined a new problem named “variable-shot adaptation for incremental meta-learning”. For its formulation, I have the following questions. (a) The system receives one data point at each step (Section 3, Page 3). However, for most incremental learning systems, a batch of training samples arrives at each time, e.g., in [A] and [B]. I think the latter one is more realistic as the training samples in vision systems aren’t often collected one by one. So why do you use a one-by-one setting instead of the common setting used in [A] and [B]? (b) In Section 3, for the Regret_T, you use theta_t instead of theta_T. It means the system is only evaluated on the current task, and you don’t care whether the model forgets the previous knowledge or not. Instead, most existing class-incremental learning and continual learning methods [A, B, I] evaluated the last model theta_T on all previous tasks, i.e., f_t(U_t(theta_T, alpha, min{s, M})) for t=1,...,T. It is commonly agreed that an incremental learning system should have the ability to retain the knowledge for all previous tasks instead of only the current one. If the proposed system is used for learning 100 different tasks, it will need to store 100 different models, which is not realistic given the memory budget in incremental learning. Again, my question is why did this paper choose this different setting (instead of the commonly agreed/used one in related works)?

2- About the baselines. This paper used MAML and FTML as the baselines. It claimed that these baselines don’t work well on variable-shot cases, e.g., can not meta-train different initialization weights for different shot numbers using MAML. While, if so, why not use metric-based methods like [C] and [D]? These methods have proved to be very effective in recent few-shot learning papers [E] and incremental learning papers [B]. Besides, it can be definitely applied to variable-shot learning cases (and in a direct manner).

3- About the novelty. The idea of meta-learning the base learning rates is not novel. It has been widely applied in many related papers, e.g., [F], [G], and [H]. [F] meta-learns base-learning rates for all base-learner parameters. [G] meta-learns layer-wise learning rates. [H] meta-learns a deep model to generate task-specific base learning rates. As a summary, the contribution of this submission is incremental, as it only simplifies the original MAML-VL to MAML-VS for the variable-shot settings.

[A] Tao, Xiaoyu, et al. "Few-Shot Class-Incremental Learning." CVPR 2020.

[B] Rebuffi, Sylvestre-Alvise, et al. "icarl: Incremental classifier and representation learning." CVPR 2017.

[C] Snell, Jake, et al. "Prototypical networks for few-shot learning." NeurIPS 2017.

[D] Vinyals, Oriol, et al. "Matching networks for one shot learning." NeruIPS 2016.

[E] Chen, Yinbo, et al. "A new meta-baseline for few-shot learning." arXiv preprint arXiv:2003.04390 (2020).

[F] Li, Zhenguo, et al. "Meta-sgd: Learning to learn quickly for few-shot learning." arXiv preprint arXiv:1707.09835 (2017).

[G] Antoniou, Antreas, et al. "How to train your MAML." ICLR 2019.

[H] Liu, Yaoyao, et al. "An Ensemble of Epoch-wise Empirical Bayes for Few-shot Learning." ECCV 2020.

[I] Li, Zhizhong and Derek Hoiem. "Learning without forgetting." IEEE TPAMI 2017.

---

> ### Author Response · Authors · 2020-11-17
> **Author Response (Part 2 of 2): Response to Con 1**
>
> **Con 1(a): “The system receives one data point at each step (Section 3, Page 3). However, for most incremental learning systems, a batch of training samples arrives at each time, e.g., in [A] and [B]. I think the latter one is more realistic as the training samples in vision systems aren’t often collected one by one. So why do you use a one-by-one setting instead of the common setting used in [A] and [B]?”**
>
> In our implementation, as mentioned in Appendix E, we do add a small batch of data every a few steps (e.g. for incremental Rainbow MNIST, we add a batch of 4 datapoints every 10 steps). Hence, our setting already resembles [A] and [B]. The description in Section 3 is the extreme case, and we have clarified this detail of our problem setting in Section 3 in the revised version. We do want to point out that since data collection and labeling are expensive in the real world, considering a small batch of data arriving at each step is realistic as discussed in Section 1. We have made the discussion more prominent in the revised version.
>
> **Con 1(b): “why did this paper choose this different setting (instead of the commonly agreed/used one in related works)?”**
>
> The difference between our setting and [A, B, I] is that we adopt the online learning set-up from [1, 2, 3] and [A, B, I] follows continual learning set-up. Both are valid settings that apply to different problem domains. We have modified the main text in Section 3 to make it clear that our method adopts the online learning setting.
>
> [1] Hannan, James. "Approximation to Bayes risk in repeated play." Contributions to the Theory of Games 3 (1957): 97-139.
> [2] Kalai, Adam, and Santosh Vempala. "Efficient algorithms for online decision problems." Journal of Computer and System Sciences 71.3 (2005): 291-307.
> [3] Finn, C., Rajeswaran, A., Kakade, S., & Levine, S. (2019). Online meta-learning. arXiv preprint arXiv:1902.08438.

---

> > ### Comment · AnonReviewer4 · 2020-11-23
> > **The feedback partially addressed my concerns**
> >
> > Thanks for the helpful feedback from the authors. The feedback partially addressed my concerns so I decide to upgrade my score from 4 to 5. However, there are still some claims which are not so convincing for me.
> >
> > For Con 1: the revised paper is clearer and it addresses my concern on how many data are used for each step. As the authors say in the response to Con 1(b), they adapt “online learning set up” instead of “continual learning/incremental learning set up”. However, the title of this paper is “incremental meta-learning”. So is that better to use “online meta-learning” instead (as [3])? Besides, as “variable-shot adaption” is a new task proposed by the authors, I think it would be better and fairer to also apply a similar set-up to [A, B, I] for the comparison.
> >
> > For Con 3: the authors claim that “per-shot scaling factor on the learning rate while [F, G, H] learn parameter/task-specific learning rates”. However, the contribution for “per-shot scaling factor on the learning rate” is straightforward and incremental. R1 also pointed out that “the proposed solutions build upon MAML and FTML, which seems incremental”. I think the feedback for these two questions can not convince me about the novelty of the proposed method. It looks like a varying-shot version of Meta-SGD. This point is still my main concern.
> >
> > [3] Finn, C., Rajeswaran, A., Kakade, S., & Levine, S. (2019). Online meta-learning. arXiv preprint arXiv:1902.08438.

---

> > > ### Author Response · Authors · 2020-11-23
> > > **Thanks for the follow-up! Addressing remaining concerns**
> > >
> > > We thank R4 for the response!
> > >
> > > **“So is that better to use “online meta-learning” instead (as [3])?”**
> > >
> > > Thanks for the suggestion! We have changed the title to “Variable-Shot Adaptation for Online Meta-Learning” and our problem setting to “online incremental meta-learning” in the revised version with changes highlighted in blue to make our setting more clear.
> > >
> > > **“Besides, as “variable-shot adaption” is a new task proposed by the authors, I think it would be better and fairer to also apply a similar set-up to [A, B, I] for the comparison.”**
> > >
> > > We have included a comparison to A-GEM (Chaudhry et al. ICLR ‘19), a widely used continual learning method that addresses catastrophic forgetting similar to [A,B,I], on the incremental Rainbow MNIST dataset in the response to R2. We also added this new experiment in Table 3 in the revised version. We reiterate the results here:
> > >
> > > Cumulative regret (lower is better):
> > > A-GEM: $14292.19 \pm 201.72$
> > > FTML-VS (ours): $\mathbf{4484.70} \pm 113.83$ (about 3x lower)
> > >
> > > **FTML-VS significantly outperforms A-GEM by a large margin**, suggesting that our theoretically-motivated per-shot scaling rule is important for variable-shot adaptation compared to prior continual learning methods that focus on minimizing negative backward transfer.
> > >
> > > **“The contribution for “per-shot scaling factor on the learning rate” is straightforward and incremental. R1 also pointed out that ‘the proposed solutions build upon MAML and FTML, which seems incremental’.”**
> > >
> > > While the modification we propose to the learning rule is simple (which should be a good thing), it improves substantially over prior methods, including FTML, ProtoNet and meta-SGD as shown in the results in the previous response and Table 2 and Table 4 in the revised paper. The value of a contribution in this context should be determined by how important it is for good performance, not how difficult it is to implement. Our derivation for the variable-shot learning rate is theoretically well-motivated, and performs well in practice. The other reviewers appear to agree that this merits publication.

---

> ### Author Response · Authors · 2020-11-17
> **Author Response (Part 1 of 2): Response to Con 2 and 3 with new experiments**
>
> Thank you for your review! We want to clarify that the setting of our incremental meta-learning does have a small batch of data arrive every few time steps and our method does not forget the knowledge of previous tasks. Moreover, we have added comparisons to prototypical networks, one of the widely used metric-based methods, on the incremental Rainbow MNIST dataset in Table 4. Finally, we also compared our method to other methods that learn the learning rates in Table 2. All changes in the revised version are highlighted in blue. Please let us know if you have any further suggestions or requests, or if we have addressed all of the issues. In the following, we reply to your specific comments.
>
> **Con 2: “While, if so, why not use metric-based methods like [C] and [D]? These methods have proved to be very effective in recent few-shot learning papers [E] and incremental learning papers [B]. Besides, it can be definitely applied to variable-shot learning cases (and in a direct manner).”**
>
> We compare our method to prototypical networks [C], one of the most popular metric-based approaches, by adapting [C] to the online incremental setting. Due to the mutually exclusive set-up, metric-based methods ([C] and [D]) are not directly applicable since they require at least one shot per class whereas the total number of support datapoints could be much less than the total number of classes in the online incremental meta-learning setting. To account for this, when the data of a certain class of the current task are not available, we sample the data of the class from previous tasks. For zero-shot generalization, we use data from previous tasks to compute the prototypes. We compare the adapted incremental prototypical networks to FTML-VS on the incremental Rainbow MNIST dataset using the data generating scheme described above, and the results are as follows:
>
> Incremental Rainbow MNIST:
> Incremental prototypical networks: $34812.7 \pm 6197.7$
> FTML-VS (ours): $\mathbf{19207.7} \pm 282.4$ (about 2x lower)
>
> FTML-VS outperforms incremental prototypical networks by a large margin. We have added this new experiment to Table 4 in the revised version.
>
> **Con 3: “The idea of meta-learning the base learning rates is not novel. It has been widely applied in many related papers, e.g., [F], [G], and [H]. [F] meta-learns base-learning rates for all base-learner parameters. [G] meta-learns layer-wise learning rates. [H] meta-learns a deep model to generate task-specific base learning rates.”**
>
> Our methods are technically orthogonal to the methods that use learned learning rates [F, G, H] since we learn per-shot scaling factor on the learning rate while [F, G, H] learn parameter/task-specific learning rates. We also conducted a comparison between FTML-VS and FTML+meta-SGD [F] on the incremental Rainbow MNIST, incremental Contextual MiniImagenet and incremental Pose Prediction datasets. The results are:
>
> Incremental Rainbow MNIST:
> FTML+Meta-SGD: $4699.0 \pm 212.0$
> FTML-VS (ours): $\mathbf{4484.7} \pm 113.8$
>
> Incremental Contextual MiniImagenet:
> FTML+Meta-SGD: $1027.3 \pm 35.8$
> FTML-VS (ours): $\mathbf{1020.0} \pm 40.8$
>
> Incremental Pose Prediction:
> FTML+Meta-SGD: $125.0 \pm 9.4$
> FTML-VS (ours): $\mathbf{119.1} \pm 3.2$
>
> FTML-VS outperforms meta-SGD in all three incremental meta-learning problems, suggesting the importance of our theoretically motivated rule of selecting learning rates that can handle variable shots. Note that we can technically also combine meta-SGD and FTML-VS to further improve performance since the two methods are complementary. We have added this new experiment to Table 2 in the revised version.
>
> The remaining answers to Con 1 are in the next post.

---

> ### Author Response · Authors · 2020-11-20
> **Request for Discussion**
>
> Dear Reviewer 4,
>
> We believe that we have addressed all of you concerns raised in the your review in our response and the revised paper. Could you please let us know if you have any additional concerns or questions? We would be happy to provide further revisions or experiments to address any remaining issues, and would appreciate a response from you on the points that we raised before the end of the discussion period on Tuesday.

---

### Official Review · AnonReviewer2 · 2020-10-27

**Rating:** 6
**Confidence:** 3

**Review:**

## Summary

The authors propose a meta-learning algorithm for online incremental settings where not only the tasks but also the points belonging to each task arrive in a sequential order. In order to effectively minimise the total amount of supervision required by the method (both for meta-training and learning each new tasks), the author extend a previous K-shots learning method to be able to deal with the variable-shots learning scenario that naturally arise when considering the sequential order in which the task datapoints are observed. In the experimental section they compare with standard meta-learning methods in a variable-shots task both offline and online. They also show that in the online case that their method is more efficient than empirical risk minimization.

## Comments

The paper is well motivated and the writing is clear although the mathematical notation could be improved.

The key observation of the paper (which is quite interesting) is that in many real world scenario tasks and datapoints arrive in sequential order, and in that case, it make sense to treat the problem as a variable shot learning problem where the aim is to minimize the total amount of supervision. Based on this observation, the authors proposal is very simple: to modify standard meta-learning algorithms (e.g. MAML offline and FTML online) by rescaling the learning rate in the inner loop according to the variable mini-batch size. The relation between the learning rate and the mini-batch size is given by a function parameterized by two additional parameters, where the functional form is derived in theorem 1. Despite its simplicity, it seems to improve the performance especially in the online setting.

As for the experimental section, the offline experiments seems a bit artificial and it is not very clear what the application would be. Also, it is unclear why the results for MAML-VL do not appear in the main manuscript, specially since it seems to perform better. On the other hand, the online experiments are more compelling and demonstrate improvement with respect to the baseline. Here I am missing a discussion about the sensitivity of the algorithm to the hyper-parameter M. Finally, while the problem tackled in the paper is different from the online meta-learning with online algorithm in the inner loop (as the authors point out in the related work) it would be interesting to add to the experimental section some baselines from these family of methods since they can be used to solve the problem setting described in the paper.

## Minors

* In the preliminaries a task is defined as a dataset, while I think it should be defined as a distribution from which a dataset is sampled by iid sampling a set of examples.
* A task dataset is defined as \mathcal{D}=\{x_i, y_i\} implying that x_i represent the covariates of all the examples in the dataset. However, later on at the beginning of page 3 x_j is used as a single datapoint. I would recommend using x_{ij} to avoid ambiguities.
* At the end of section 2, in the equation for \theta_{t+1}, I think the minimization should be over \theta (not over \theta_j). Same in equation (2).
* At beginning of section 4.3 it should be added what VL stands for
* Appendix A: VS-Meta-Update receives "s" as input. However, "s" does not appear inside the function

---

> ### Author Response · Authors · 2020-11-17
> **Author Response**
>
> Thank you for your review! We have clarified the settings of our offline experiments in Section 7.1, added ablation studies on the hyperparameter M using the incremental Rainbow MNIST dataset in Appendix G, and compared to a popular continual learning method A-GEM in Table 3. All changes in the revised version are highlighted in blue. Please let us know if you have any further suggestions or requests, or if we have addressed all of the issues. In the following, we reply to your specific comments.
>
> **“The offline experiments seems a bit artificial and it is not very clear what the application would be.”**
>
> Our main results are the online experiments,  but the offline experiments are designed to show that our variable-shot meta-learning method can help improve the performance when evaluated with variable shots with all tasks available. It can be viewed as a sanity check for the online experiments. We’ve added this clarification to the modified version. Moreover, the offline experiments focus on meta-learning problems where the tasks are not mutually exclusive, which is often more realistic than the meta-learning benchmarks where tasks are artificially made to be mutually exclusive [1].
>
> [1] Yin, Mingzhang, et al. "Meta-learning without memorization." arXiv preprint arXiv:1912.03820 (2019).
>
> **“It is unclear why the results for MAML-VL do not appear in the main manuscript, specially since it seems to perform better.”**
>
> Thanks for pointing this out! We have moved the results for MAML-VL to the main text.
>
> **“Here I am missing a discussion about the sensitivity of the algorithm to the hyper-parameter M.”**
>
> We have added an ablation study on the sensitivity of M on the incremental Rainbow MNIST dataset to the revised version. The results are as follows:
>
> Incremental Rainbow MNIST (M=10):
> FTML: $7710.0 \pm 769.8$
> FTML-VS (ours): $\mathbf{5643.3} \pm 149.0$
>
> Incremental Rainbow MNIST (M=20):
> FTML: $4804.2 \pm 302.8$
> FTML-VS (ours): $\mathbf{4484.7} \pm 133.8$
>
> Incremental Rainbow MNIST (M=30):
> FTML: $4250.7 \pm 253.6$
> FTML-VS (ours): $\mathbf{3969.0} \pm 203.7$
>
> As shown in the results, as M increases, the cumulative regret of both FTML and FTML-VS decreases since learning becomes easier with larger numbers of shots. Meanwhile, FTML-VS attains better performance compared to FTML with different values of M and the performance gap becomes larger as M decreases, suggesting that our theoretically motivated learning rate scaling rule based on the number of shots is important for different values of the maximum number of shots and is particularly effective when the maximum number of shots is small. We added this ablation study to Appendix G.
>
> **“While the problem tackled in the paper is different from the online meta-learning with online algorithm in the inner loop (as the authors point out in the related work), It would be interesting to add to the experimental section some baselines from these family of methods since they can be used to solve the problem setting described in the paper.”**
>
> Methods that apply online meta-learning with an online algorithm in the inner loop consider the setting where all meta-training tasks are available in batch ahead of time, while we assume that we have incremental access to tasks and data seen within each observed task. Hence, these methods are not applicable to our problem settings. We will clarify this in the related work setting to make this more clear in the revised version.
>
> Meanwhile, since we cannot apply methods that study continual learning in the inner loop of meta-learning to our experiments, we consider comparisons to continual learning methods, i.e. the inner loop of methods discussed above, which is applicable to our setting. We conducted a comparison to A-GEM (Chaudhry et al. ICLR ‘19), a widely used continual learning method, on the incremental Rainbow MNIST dataset. We ran A-GEM and FTML-VS for three random seeds, following the protocol in our paper. The results are:
>
> Cumulative regret (lower is better):
> A-GEM: $14292.19 \pm 201.72$
> FTML-VS (ours): $\mathbf{4484.70} \pm 113.83$ (about 3x lower)
>
> **FTML-VS outperforms A-GEM by a large margin.** This result is unsurprising, as prior continual learning works focus primarily on minimizing negative backward transfer and compute considerations, as opposed to our goal of accelerating forward transfer through meta-learning. We have added this new experiment to Table 3 in the revised version.
>
> **Minors**
>
> Thank you for catching those! We have fixed them in the revised version.

---

> ### Author Response · Authors · 2020-11-20
> **Request for Discussion**
>
> Dear Reviewer 2,
>
> We believe that we have addressed all of you concerns raised in the your review in our response and the revised paper. Could you please let us know if you have any additional concerns or questions? We would be happy to provide further revisions or experiments to address any remaining issues, and would appreciate a response from you on the points that we raised before the end of the discussion period on Tuesday.

---

### Official Review · AnonReviewer1 · 2020-10-29
**An Interesting work with meta-learned learning rate.**

**Rating:** 6
**Confidence:** 4

**Review:**

The authors aim to tackle the meta learning problem in online settings, with both training and testing samples received in a streaming fashion. The authors proposed both offline and online methods, MAML-VS and FTML-VS, built upon MAML and FTML, respectively. The key contribution is to learn meta-learning rates besides the initial network parameters. Overall, the proposed approach sounds.

The contributions of this paper are as follows:

1.	The proposed online and offline meta learning algorithms aim to tackle the few-shot meta-learning problems with variable amounts of data received in a stream.

2.	The paper shows the limitation of MAML and FTML in the online setting. It also provides theoretical results on the meta learning rates in Theorem 1.

3.	The authors have done extensive experiments to evaluate the effectiveness of the proposed approaches in various datasets, e.g., RAINBOW MNIST, Contextual Mini-Imagenet, etc.

To the current status of the paper, I have a few concerns below.

First, the proposed solutions build upon MAML and FTML, which seems incremental. The differences include (i) a modification on the objective function in eq. (1) for the streaming/online setting, and (ii) a meta-learned learning rate for better model convergence property.

Second, in table 1 (offline setting), it shows that MAML-VS outperforms baselines in fewer-shot cases with Rainbow MNIST, and in 10/20-shot cases with Contextual imageNet. It is unclear how to explain this inconsistency. Overall, the proposed MAML-VS does not perform better than baselines in offline settings.

Moreover, it would be interesting to also show how the proposed approach works in reinforcement learning settings.

Minors:

In the pseudo-code in section5.2 (line 3), it should be $j=t$.

---

> ### Author Response · Authors · 2020-11-17
> **Author Response**
>
> Thank you for your review! We have addressed all the raised concerns in a revised version of the paper. Please let us know if you have any further suggestions or requests, or if we have addressed all of the issues. In the following, we reply to your specific comments.
>
> **“It is unclear how to explain this inconsistency. Overall, the proposed MAML-VS does not perform better than baselines in offline settings”**
>
> Compared to Contextual MiniImagenet, Rainbow MNIST is a much easier task. Therefore 10 or 20 shot adaptation is sufficient for any algorithms to perform well. Our algorithm performs comparably to the baselines in 10/20 shot settings while attaining better performances in 0/1 shot adaptation. For Contextual MiniImagenet, since it is a much more challenging problem with large task space, 0/1/10/20-shot adaptation is hard for all tasks. MAML-VS achieves better 10/20-shot adaptation performances while maintaining comparable to the best performance of other methods in the 0/1-shot adaptation setting.  In this paper however, we are mostly focusing on the online incremental setting, so we include the offline experiments to demonstrate that our variable shot method is not exclusive to the online setting. Moreover, since in the online incremental setting, performing well with all numbers of shots is important, our offline experiments show that our algorithm could be much more effective than baselines in the online incremental setting.
>
> **“The proposed solutions build upon MAML and FTML, which seems incremental. The differences include (i) a modification on the objective function in eq. (1) for the streaming/online setting, and (ii) a meta-learned learning rate for better model convergence property”**
>
> Our method, which is a rule that automatically selects the learning rate based on the number of shots, is theoretically motivated and empirically effective especially in the online incremental setting. Though it is a simple modification of the objective, the theoretical soundness and empirical performance should still make the method a novel contribution.
>
> **“Moreover, it would be interesting to also show how the proposed approach works in reinforcement learning settings”**
>
> This is a great suggestion, and we hope to pursue this direction in future works.
>
> **Minor**
>
> Thank you! We have fixed it in the revised version.

---

> ### Author Response · Authors · 2020-11-20
> **Request for Discussion**
>
> Dear Reviewer 1,
>
> We believe that we have addressed all of you concerns raised in the your review in our response and the revised paper. Could you please let us know if you have any additional concerns or questions? We would be happy to provide further revisions or experiments to address any remaining issues, and would appreciate a response from you on the points that we raised before the end of the discussion period on Tuesday.

---

### Official Review · AnonReviewer3 · 2020-11-06
**A well-written paper and potentially a new setup for few-shot learning**

**Rating:** 6
**Confidence:** 4

**Review:**

## Summary
Following Finn et al. 2019, this paper aims at solving incremental meta-learning problems. A new setup is proposed as illustrated in Fig 1, which is motivated by learning a model that is capable of generalizing to a new task with decreasing number of shots. For the offline and online settings of incremental meta-learning, this paper proposes the MAML-VS algorithm and the FTML-VS algorithm, which are based on Finn et al. 2017 and Finn et al. 2019. Offline and online experiments are conducted on 4 benchmarks showing good performance comparing to baselines, where the Contextual MiniImageNet is a new a dataset proposed by this paper.

## Contributions
1. Proposed a practical setup for few-shot learning when the tasks are non-mutually exclusive.
2. Proposed the learning rate scaling method for variable shots.

## Issues
1. It is not clear when and why zero-shot should work if no information about that task is revealed. More details should be elaborated on this point.
2. In online meta-learning, each task as a different loss function, while in Theorem 1, the result is based on the assumption that all loss functions are the same. Can your result as well as the learning rate scaling method extend to the general setting?
3. The performance of MAML-VS on Omniglot is worse than MAML. Can you try MAML-VS on other mutually exclusive datasets such as MiniImageNet?

---

> ### Author Response · Authors · 2020-11-17
> **Author Response**
>
> Thank you for your review! We have addressed all the raised concerns in a revised version of the paper. Please let us know if you have any further suggestions or requests, or if we have addressed all of the issues. In the following, we reply to your specific comments.
>
> **“It is not clear when and why zero-shot should work if no information about that task is revealed. More details should be elaborated on this point”**
>
> In this paper the tasks are assumed to be mutually non-exclusive, which means that potentially the tasks can be solved with one single non-adaptive agent. This setting is often more realistic than the meta-learning benchmarks where tasks are artificially made to be mutually exclusive [1]. We will elaborate on this point in Section 7 to make it more clear.
>
> [1] Yin, Mingzhang, et al. "Meta-learning without memorization." arXiv preprint arXiv:1912.03820 (2019).
>
> **“In online meta-learning, each task as a different loss function, while in Theorem 1, the result is based on the assumption that all loss functions are the same. Can your result as well as the learning rate scaling method extend to the general setting?”**
>
> In this paper as well as in most prior works of online meta-learning, the loss function for all tasks is the same and only the data is changing. Therefore Theorem 1 applies to the Rainbow MNIST and Contextual ImageNet experiments we include in this paper. In the general setting where the parameterization of learning rate is varying, one can simply multiply a different base learning rate for each loss parameterization, which is orthogonal to our method.
>
> **“The performance of MAML-VS on Omniglot is worse than MAML. Can you try MAML-VS on other mutually exclusive datasets such as MiniImagenet?”**
>
> On mutually exclusive datasets, we provide the number of shots per class, which results in a much larger number of shots compared to the number used in non-mutually exclusive datasets. This difference could explain why MAML-VS does not outperform MAML in the mutually exclusive setting. Also, note that the main contribution of this paper is the online incremental experiments and the offline experiments on mutually exclusive datasets are used to show that our method is not exclusive to non-mutually exclusive datasets.
>
> We also compared MAML-VS to MAML on MiniImagenet. Here are the results:
>
> MiniImagenet (5-way 1-shot):
> MAML: **46.66%**
> MAML-VS (ours): 46.06%
>
> MiniImagenet (5-way 2-shot):
> MAML: 48.12%
> MAML-VS (ours): **48.3%**
>
> MiniImagenet (5-way 5-shot):
> MAML: **52.34%**
> MAML-VS (ours): 52.17%
>
> MAML-VS performs similarly to MAML, which is expected due to the reason mentioned above.

---

> > ### Comment · AnonReviewer3 · 2020-11-24
> > **After rebuttal**
> >
> > Thanks to the authors for answering the questions! The rebuttal has addressed some of my concerns.
> >
> > I however would like to keep my previous rate because now I think this work may have fewer applications than I was expecting. It seems the main algorithm is essentially designed for non-mutually exclusive online few-shot learning, which wasn't clearly explained in the methodology section (delayed until very late in Sec 7). As commented by the authors, the variable-shot extension of MAML has a fundamental issue which may be even worse than MAML in the offline setting.
> >
> > Besides, as pointed out by other reviewers, the design is tight to optimization based meta-learning making it difficult to be extended to a broader family of meta-learning algorithms.

---

> > > ### Author Response · Authors · 2020-11-24
> > > **Thanks for the response! Addressing remaining concerns**
> > >
> > > We thank R3 for the response! We would like to address your concerns as follows.
> > >
> > > **“It seems the main algorithm is essentially designed for non-mutually exclusive online few-shot learning … the variable-shot extension of MAML has a fundamental issue which may be even worse than MAML in the offline setting”**
> > >
> > > We would like to clarify that our method (MAML-VL and MAML-VS) is comparable to MAML in the mutually-exclusive setting with a much higher number of shots (at least one shot per class) as shown in Table 5 and the miniImagenet results posted above and better than MAML in the non-mutually exclusive setting with a lower number of shots as shown in Table 1. Thus, our method does work in both the offline mutually-exclusive and the offline non-mutually exclusive settings. They are just not the main settings we are trying to improve on. We believe this is not a “fundamental issue” with our method. We will make this clarification more clear and appear more prominently in a revised version.
> > >
> > > **“the design is tight to optimization based meta-learning making it difficult to be extended to a broader family of meta-learning algorithms”**
> > >
> > > As shown in the response to R4 and Table 4 in the revised version (highlighted in blue), our setting can be extended to non-parametric meta-learning algorithms and we compared our method FTML-VS to the adapted incremental prototypical networks (Snell, Jake, et al., NeurIPS 2017) on the incremental Rainbow MNIST dataset and found that FTML-VS significantly outperformed incremental prototypical networks. We reiterate this experiment here:
> > >
> > > Incremental Rainbow MNIST:
> > > Incremental prototypical networks: $34812.7 \pm 6197.7$
> > > FTML-VS (ours): $\mathbf{19207.7} \pm 282.4$ (about 2x lower)

---

> > > > ### Comment · AnonReviewer3 · 2020-11-24
> > > > **After rebuttal II**
> > > >
> > > > Thanks for the quick response!
> > > >
> > > > I still have two concerns:
> > > >
> > > > 1. From table 1, 5 and mutually exclusive MiniImageNet, the baseline MAML works pretty well under variable-shot setting. Doesn't that suggest the learning rate scaling method is a minor thing?
> > > >
> > > > 2. "As shown in the response to R4 and Table 4 in the revised version (highlighted in blue), our setting can be extended to non-parametric meta-learning algorithms"
> > > > I still don't see how to extend the learning rate scaling method to non-parametric methods.

---

> > > > > ### Author Response · Authors · 2020-11-25
> > > > > **Thanks for the fast response! Further clarifications**
> > > > >
> > > > > Thank you for your fast response! We would like to make further clarifications regarding your concerns as follows.
> > > > >
> > > > > **Point 1: “From table 1, 5 and mutually exclusive MiniImageNet, the baseline MAML works pretty well under variable-shot setting. Doesn't that suggest the learning rate scaling method is a minor thing?”**
> > > > >
> > > > > We clarify that our main results are the online experiments, but the offline experiments are designed to show that our variable-shot meta-learning method can help improve the performance when evaluated with variable shots with all tasks available. It can be viewed as a sanity check for the online experiments. We’ve added this clarification to the modified version in Section 7 as you mentioned and will make it appear more prominently in a revised version. Regarding the results in Table 1, our algorithm performs comparably to MAML in 10/20 shot settings while attaining better performances in 0/1 shot adaptation on the Rainbow MNIST. Moreover, our method outperforms MAML in all of variable-shot adaptation settings on the Contextual MiniImagenet. Since in the online incremental setting, performing well with all numbers of shots is important, our offline experiments show that our algorithm could be much more effective than baselines in the online incremental setting, which is later confirmed in the online experiments. For mutually exclusive results in Table 5 and the added MiniImagenet experiments, we just want to show that our method is not exclusive to the non-mutually exclusive settings and does not make performance much worse in that setting. Achieving the best performance on the offline mutually exclusive datasets is not the focus of our method. We will also make this clarification appear more prominently in a revised version of the paper.
> > > > >
> > > > > **Point 2: “I still don't see how to extend the learning rate scaling method to non-parametric methods.”**
> > > > >
> > > > > Thanks for pointing it out! We acknowledge that our learning rate scaling method is limited to optimization-based meta-learning methods, but we want to point out that we also compare our method to the non-parametric methods that naturally extends to the online incremental setting and show that our method significantly outperforms the non-parametric approaches.

---

> ### Author Response · Authors · 2020-11-20
> **Request for Discussion**
>
> Dear Reviewer 3,
>
> We believe that we have addressed all of you concerns raised in the your review in our response and the revised paper. Could you please let us know if you have any additional concerns or questions? We would be happy to provide further revisions or experiments to address any remaining issues, and would appreciate a response from you on the points that we raised before the end of the discussion period on Tuesday.

---

### Author Response · Authors · 2020-11-23
**Title change**

As mentioned in the response to R4, we now changed the title to **"Variable-Shot Adaptation for Online Meta-Learning"** to make it clear that our problem setting is online learning rather than continual learning or incremental learning. The original one was "Variable-Shot Adaptation for Incremental Meta-Learning".

---

### Decision · Program_Chairs · 2021-01-07
**Final Decision**

**Decision:**

Reject

**Comment:**

For meta-learning with variable shot, this paper proposes a method for adapting the learning rate by a function of the number of training examples. The functional form is theoretically derived, and the method is simple and effective. However, meta-learning methods that adapt learning rates have been proposed, and the novelty is not high enough.